# Physical Activity during the First Lockdown of the COVID-19 Pandemic: Investigating the Reliance on Digital Technologies, Perceived Benefits, Barriers and the Impact of Affect

**DOI:** 10.3390/ijerph18115555

**Published:** 2021-05-22

**Authors:** Michelle Symons, Carmem Meira Cunha, Karolien Poels, Heidi Vandebosch, Nathalie Dens, Clara Alida Cutello

**Affiliations:** 1Department of Communication Studies, Faculty of Social Sciences, University of Antwerp, 2000 Antwerp, Belgium; karolien.poels@uantwerpen.be (K.P.); heidi.vandebosch@uantwerpen.be (H.V.); 2Department of Marketing, Faculty of Business Economics, University of Antwerp, 2000 Antwerp, Belgium; carmem.meiracunha@uantwerpen.be (C.M.C.); nathalie.dens@uantwerpen.be (N.D.); clara.cutello@uantwerpen.be (C.A.C.)

**Keywords:** physical activity, COVID-19 lockdown, digital support for exercise, benefits, barriers, affect

## Abstract

The measures to fight the spread of the COVID-19 pandemic have been concentrated on inviting people to stay at home. This has reduced opportunities to exercise while also shedding some light on the importance of physical health. Based on an online survey, this paper investigated physical activity behaviours of a Belgians sample (*n* = 427) during the lockdown period between the end of May 2020 and the beginning of June 2020 and found that, during this period, the gap between sufficiently and insufficiently active individuals widened even more. This paper analysed important moderators of physical activity behaviours, such as barriers and benefits to exercise, digital support used to exercise, and individuals’ emotional well-being. Descriptive analysis and analyses of variance indicated that, generally, individuals significantly increased their engagement in exercise, especially light- and moderate-intensity activities, mostly accepted the listed benefits but refused the listed barriers, increased their engagement in digital support and did not score high on any affective measures. A comparison between sufficiently active and insufficiently active individuals during the lockdown showed that the former engaged even more in physical activity, whereas the latter exercised equally (i.e., not enough) or even less compared to before the lockdown. By means of a logistic regression, five key factors of belonging to the sufficiently active group were revealed and discussed. Practical implications for government and policies are reviewed.

## 1. Introduction

### 1.1. Coronavirus

Due to the outbreak of the COVID-19 pandemic, governments all over the World took protective measures to contain the spread of the virus. Overall, these measures have been highly concentrated on inviting people to self-isolate and stay home [1]. Between March and June (hereinafter referred to as first lockdown), in most Western European countries, governments closed several public places, imposed bans on various events that required social gatherings, and solicited residents to physically distance themselves from others [2]). Expansion of these measures often resulted in (partial) lockdowns, which was also the case for Belgium, where outings were only allowed when essential, and routine activities had to be adapted and to be performed at home. In addition, penalties would force individuals to abide by the rules [3].

### 1.2. Physical Activity

Despite implementing these restricting measures, governments kept advising their residents to continue to stay active, emphasizing (outdoor) physical activity [4,5]. Physical activity is defined as “*any bodily movement produced by skeletal muscles that require energy expenditure*” [6]. The World Health Organization distinguishes three levels of physical activity intensity: light-intensity, moderate-intensity and vigorous-intensity [7,8]. Light intensity exercise refers to activities where individuals do not breath any faster, their heartrate remains normal, and they can talk normally [9]. Moderate intensity exercise refers to activities that are performed at 3.0 to 5.9 times the intensity of rest and can demand between 50% and 60% of the individual’s personal capacity for exercise. In comparison with light intensity exercise, individuals will breathe faster, their heartbeat will increase, but they will not be out of breath [10,11]. Vigorous-intensity physical activities are performed at 6.0 or more times the intensity of rest and required an individual’s personal capacity of 70% or 80% [11]. Concretely, this means that individuals will breathe much faster (also referred to as aerobic’ movements), their heartbeat will increase significantly, and they will be out of breath [12]. The (inter)national recommendations for physical activity are dependable on age [7,13]. All ages should engage in light intensity exercise as much as possible. Adults, aged 18 to 64 years old, should either engage in 150 min of moderate exercise or 75 min of vigorous exercise per week, with strengthening exercises performed two or more days a week. A person is considered insufficiently active or inactive when (s)he fails to meet these recommendations for physical activity [14]. Although the benefits of physical activity concerning physical health are often related to higher intensity workouts, the opposite is true for mental health benefits. Light to moderate intense activities, such as waking, are presumed to be associated with improvements in moods, cognitive functioning, overall quality of life and reductions in anxiety [15]. For example, an experimental study found that accumulating 10,000 steps a day (or more) can decrease negative moods, including anxiety, depression and confusion [16]. In addition, regular exercise has been found to prevent health-related diseases [17], and past research has found that people who engage in regular exercise of moderate intensity report less symptoms of upper respiratory tract infections [18,19]. Therefore, during a pandemic that pronominally involves the respiratory tract, physical activity becomes of salient importance.

### 1.3. Benefits and Barriers

Because of the importance of physical activity in general and during the spread of the coronavirus, this pandemic shifts existing health benefits of exercising, such as physical but also social benefits [20] to the forefront of people’s attention during the lockdown [21] and may have even allowed for some pre-existing barriers to be removed, such as not having a lot of time for physical activity or having the need to move to a sports centre. In addition, more avid sportsmen have the opportunity to fully recover from competition stress, injuries and accumulated loads of overtraining [22,23]. Hence, this pandemic can be considered as a momentum of reflection on our existing health behaviours, which has the potential to stimulate individuals to live differently than before [24,25]. Nevertheless, with the suspension of athletic programs, and the closure of fitness centres and sometimes even public parks [26], the population had to face new barriers to exercise in comparison to before the lockdown. Barriers can be defined as challenges and difficulties that negatively influence physical activity levels [27].

### 1.4. Digital Support

A way to cope with these barriers is to make use of technological support tools such as smartphone applications and physical activity trackers [28]. Technological support often involves self-tracking, which increases awareness of automated, habitual behaviours and their consequences [29]. The lockdown provides opportunities to register basic activity levels and become aware of the influence of these measures [25]. Subsequently, the need to adjust our lives to lockdown regulations resulted in moving daily activities from the offline to the online sphere [30], such as working [31], teaching [32], socially interacting [33,34] and also engaging in physical activities [35,36]. As an example, fitness clubs and fitfluencers started offering their support online by giving online classes, exercise instructions and advice [35]. This online support, that before the lockdown was mainly provided offline (e.g., sport lessons), will be referred to as mediated support.

### 1.5. Affect

Aside from providing guidance for exercise, mediated support also had the ability to be used and used as a way to fight the loneliness caused by the social isolation. Previous literature has demonstrated that loneliness is associated with increased media use to find companionships and new social interactions [37]. However, loneliness is not the only consequence of social isolation, as it has been found to correlate with many mental health problems (e.g., anxiety and depression), even in people who had never experienced mental health issues before [38]. Overall, pandemics are found to have a negative impact on the well-being of individuals by increasing negative emotions and decreasing positive emotions [39,40]. Furthermore, affect and physical activity behaviours are closely related [41,42], with positive affect being associated with concurrent and long-term health behaviours [43].

### 1.6. Objectives of the Paper

As the lockdown enhances barriers but also gives the opportunity to transform barriers into benefits, this study will assess whether individuals recognized these new benefits and barriers and how they handled this new situation in terms of physical activity behaviours. Therefore, the aim of this paper is to examine and describe individuals their physical activity behaviours during lockdown. In addition, this paper aims to investigate the reliance on digital technologies, perceived benefits, barriers and the impact of affect in relation to physical activity during lockdown. The research contained in this study can help guide government and institutions even better on how to adapt their policies, campaigns and interventions to provide support for people to stay at home and become or remain physically active.

## 2. Materials and Methods

As this study collected data after almost ten weeks of lockdown, it could capture a better picture of how individuals actually adapted their physical activity behaviours to the lockdown, rather than capturing a first glance reaction to reduced mobility. This paper will investigate individuals’ self-reported physical activity behaviours during the lockdown and compare this to their perceived physical activity behaviours before the lockdown—all measured in each of the three intensity categories of physical activity: light, moderate and vigorous [44]. Secondly, this paper will analyse specific benefits and barriers plausible to be experienced during the lockdown. Thirdly, current and intended future use of technological support tools, including mediated support, will be researched to better understand the relationship with these support systems. Furthermore, this paper will consider individuals’ affect, as this will help to further understand their motivations and psychological health. Lastly, as previous research shows that motivations and barriers differ between active and inactive individuals [45], this critical distinction will also be made in this study.

### 2.1. Sample and Procedure

This study consisted of an online questionnaire administered through the survey platform Qualtrics. Responses were gathered at the end of the first Belgian lockdown in Flanders (i.e., the largest, Dutch-speaking northern part of Belgium), between 19 May and 7 June 2020 (the Belgian government adopted more flexible measures from 8 June). Respondents in this study were recruited by means of snowball sampling (see Table 1). This sampling method allowed for a fast data collection and accommodated respondents’ willingness to participate. Out of 496 respondents that finished filling in the survey, 427 were included in the analysis—30 were excluded for absence of agreement with the data processing, 12 did not pass the attention check and 27 were not living in Belgium during the lockdown. This study followed APA Ethical Guidelines for research with human subjects and received approval from the university’s Ethics Committee for the Social Sciences and Humanities. No compensation was given for participation.

### 2.2. Measures

#### 2.2.1. Demographic Measures

After reading a brief introduction and confirming that they were older than 18 years old, respondents were asked to indicate their gender, age, living situation (with whom), and occupational status (before and during the pandemic).

#### 2.2.2. Physical Activity

Respondents were asked to self-report their exercising behaviour for activities of light, moderate and vigorous intensity during the lockdown. These questions included the number of days per week they exercised and the average duration of the workouts (e.g., Think about a ‘normal’ week during lockdown. We would like to receive some information about your light intensity physical activity behaviours. Light intensity exercise refers to activities where you do not breath any faster, your heartrate remains normal, and you can talk normally [33], such as walking, yoga and golf. Please indicate the average number of days per week (frequency) you engage in this type of activity (between 0 and 7), followed by the average time per day you engage in this type of activity (duration)). These questions were based on two validated scales: The International Physical Activity Questionnaire (IPAQ) [46] and the Godin–Shephard Leisure-Time Physical Activity Questionnaire [47] but needed some adaptations that were made by physical activity experts due to lockdown environment. To calculate the total minutes people reported to spend on each level of exercise per week frequency (e.g., 3 days a week) and duration (e.g., 30 min) measures were multiplied (e.g., 90 min of exercise per week). Additionally, respondents were asked if they were doing more or less on each type of physical activity compared to before the lockdown (in %) (e.g., please think about a normal week before the lockdown. Do you feel like your vigorous-intensity physical activities have changed compared to before the lockdown? Please indicate on the bar below: if you indicate 100%, this means that there has been no change in your light-intensity physical activities. A percentage lower than 100% means you are doing less and a percentage greater than 100% means your physical have increased).

#### 2.2.3. Benefits and Barriers

Respondents had to indicate how much they agreed (on a 5-point Likert scale with 1 = totally disagree to 5 = totally agree) with a list of benefits and barriers to exercising during lockdown with benefits consisting of 6 items, such as life improvement, physical achievements, social interactions, etc. and barriers consisting of 5 items, such as physical effort, lack of family encouragement, exercise milieu, etc.. Both benefits and barriers were measured using the psychometric evaluation of the exercise benefits and barriers scale [20], with the addition of “disruptions due to measures to fight COVID-19″ as a barrier to exercise and different time allocation due to lockdown as a benefit and barrier.

#### 2.2.4. Digital Support

Respondents were asked about how often they relied on different types of technological (e.g., smartphone application) and mediated support (e.g., online sports classes) for exercising, before and during lockdown, as well as to what extent they intended to keep using these types of support after the lockdown (all on 5-point Likert scales with 1 = never to 5 = often).

#### 2.2.5. Affect

Emotional well-being was measured with the short international PANAS scale consisting of ten items measured on 5-point Likert scales with 1 = totally disagree to 5 = totally agree [48]. In this scale, five items referred to positive affective measures such as alert and inspired and five to negative affective measures, such as hostile and scared. An overall positive and an overall negative affective measure was calculated by taking the sum of the individual items. The emotional well-being scale showed a good internal consistency, with a Cronbach’s alpha coefficient of 0.775. Additionally, anxiety was measured by the short PROMIS anxiety questionnaire (e.g., I felt anxious during lockdown) [49] and fatigue was measures by the short PROMIS fatigue questionnaire (e.g., I felt fatigue during lockdown) [50,51], both consisting of four items measured on 5-point Likert scales with 1 = totally disagree to 5 = totally agree. Both scales showed good internal consistency, with Cronbach’s alpha coefficients of 0.878 and 0.900, respectively. Lastly, fear of COVID-19 was measured with the fear of COVID scale, consisting or seven items such as, ‘I am afraid of the coronavirus’ [52]-all measured on 5-point Likert scales with 1 = totally disagree to 5 = totally agree with Cronbach’s alpha coefficients of 0.822, respectively.

## 3. Results

### 3.1. Characteristics of the Sample

The average age of the respondents was 34 years old (SD = 14.12), and 82.9% of the individuals were women. Respondents were highly educated: 42.6% had a Bachelor (Professional or Academic), 34.9% had a Master and 9.6% had a PhD degree (only 12.9% had up to secondary education). Most respondents lived either with their parents (29%) or with a partner (27.4%), and some lived with a partner and children (full-time) (13.6%) or alone (11.7%) (18.3% had other household situations). Although 76.3% of the respondents did not have children, 5.9% had one and 12.2% had two in the house during lockdown (5.6% had more). Only 12.6% of the individuals had a different household situation before the lockdown. Regarding employment, 40.7% worked full-time, 28.1% were students, and 14.3% did not work (16.9% worked more or less than part-time). The employment situation was the same before and during the lockdown for 82.9% of the respondents. Out of those whose situation has changed, most were full-time employed and became unemployed or working part-time (56.8%).

### 3.2. Physical Activity during Lockdown

We first identified potential outliers based on the Z-scores for the frequency per week and duration per day of exercise for all three levels of physical activity. All absolute values higher than 2.5 were considered as outliers, resulting in excluding 38 respondents from the analyses [53]. During lockdown, respondents engaged in physical activity between 0 and 7 days a week for light and moderate levels of physical activity and between 0 and 6 days a week for vigorous levels of physical activity (see Figure 1). They engaged most frequently in light physical activity (M = 3.82, SD = 2.18), followed by moderate exercise (M = 2.50, SD = 1.92), with vigorous physical activity being least frequent (M = 1.70, SD = 1.78). The same order applied for the duration of their exercise per session: on average, individuals engaged 49.38 min (SD = 31.45) in light-intensity activities, for 44.15 min (SD = 35.54) in moderate-intensity activities and for 29.76 min (SD = 31.44) in vigorous-intensity activities. Respondents engaged in light-intensity activities between 0 min and 150 min per session, between 0 and 180 min of moderate-intensity activities and between 0 and 130 min of vigorous activities per session (see Figure 2). Around 7.00% of respondents never performed light-intensity physical activity and 15.50% did not engage in moderate exercise at all. This percentage for vigorous-intensity physical activity was 38.87%. Only 10 people did not engage in exercise at all, regardless of intensity.

Combining frequency and duration measures revealed the total minutes people reported to spend on each level of exercise per week (see Figure 3). On average, people spent 205.86 min per week (SD = 176.71) engaging in light, 125.95 min (SD = 127.10) engaging in moderate and 82.10 min (SD = 106.78) engaging in vigorous physical activity.

#### Comparing Physical Activity Levels before and during Lockdown

A one sample *t*-test indicated that respondents scored significantly higher on light-intensity activities (M = 114.57; SD = 49.47; t(426) = 6.084; *p* < 0.001) and moderate-intensity activities (M = 111.05; SD = 52.36; t(426) = 4.363; *p* < 0.001) than the neutral value of 100 (which was indicated as ‘no change in comparison with before the lockdown’). Indicating that they significantly increased their level of light and moderate intensity physical activity. Respondents perceived their engagement in vigorous-intensity activities also as higher than before the lockdown, but this increase was not found to be significant (M = 103.84; SD = 57.38; t(426) = 1.383; *p* = 0.167).

### 3.3. Benefits and Barriers during Lockdown

The most common benefits to exercise during the lockdown were reportedly life improvement—such as better sleep and improved functioning of the body (M = 4.14; SD = 0.831) and psychological perspectives—e.g., enjoyment of exercise, posterior relaxation, reduction of stress and tension (M = 4.01; SD = 0.870), and physical performance—i.e., increased muscle strength, improved health functions, improved endurance (M = 3.88; SD = 1.001). The least indicated benefits were time allocation as an opportunity to exercise, i.e., having more time to exercise in comparison with before the lockdown or to break the monotony of the lockdown (M = 3.53; SD = 1.339), preventive health (e.g., prevention of heart attacks, prevention of high blood pressure…) (M = 3.44; SD = 1.120) and social interactions (M = 2.70; SD = 1.239) (see Figure 4). A one-sample *t*-test showed that respondents scored significantly higher on all benefits than the neutral agreement score (value of three) (t_life improvement_ (426) = 28.247, *p* < 0.001; t_psychological perspectives_ (425) = 24.049, *p* < 0.001; t_physical performance_ (426) = 8.114, *p* < 0.001; t_time allocation_ (424) = 8.116, *p* < 0.001; t_preventive health_ (423) = 8.067, *p* < 0.001) with the exception of social interactions, which scored significantly lower than the neutral agreement score (value of three) (t(424) = −4.934, *p* < 0.001).

Respondents mostly felt that the listed potential barriers were not strong barriers to exercise during the lockdown (see Figure 4). A one-sample *t*-test showed that respondents scored significantly lower on all barriers than the neutral agreement score (value of three): time allocation as a barrier—i.e., engagement in exercise takes too much time, I need this time for my family responsibilities and family relationships (M = 2.71; SD = 1.303; t(424) = −4.652; *p* < 0.001), motion disruption by COVID, which was mostly related to reduction in infrastructure and opportunities to exercise (M = 2.55; SD = 1.455; t(424) = −6.336; *p* < 0.001), physical effort—i.e., exercising is too tiring or too hard (M = 2.47; SD = 1.166; t(422) = −9.301; *p* < 0.001), lack of family encouragement (M = 2.429; SD = 1.092; t(423) = −13.474; *p* < 0.001) and sport environments being too far away from home (M = 1.85 SD = 0.964; t(423) = −24.489; *p* < 0.001).

#### Benefits and Barriers per Level of Physical Activity during Lockdown

A multivariate analysis of variance (MANOVA) with the listed benefits (live improvement, psychological perspectives, physical performance, better time allocation, preventive health and social interactions) as independent variables and the three levels of physical activity (light, moderate and vigorous) as dependent variables, revealed that better time allocation due to the lockdown (F(3, 375) = 11.607, *p* < 0.001, ηp^2^ = 0.085, Wilks’ Lambda = 0.915), physical performance (F(3, 375) = 8.914, *p* < 0.001, ηp^2^ = 0.067, Wilks’ Lambda = 0.933) and preventive health (F(3, 375) = 3.222, *p* < 0.050, ηp^2^ = 0.250, Wilks’ Lambda = 0.975) were significant predictors of overall physical activity levels. Tests of Between-Subjects Effects indicated that a better time allocation due to the lockdown was a significant predictor for all levels of physical activity (F_LPA_(1, 377) = 22.303, *p* < 0.001, ηp^2^ = 0.056; F_MPA_(1, 377) = 10.690, *p* < 0.001, ηp^2^ = 0.028 and F_VPA_(1, 377) = 11.881, *p* < 0.001, ηp^2^ = 0.031). Physical performance was only a predictor for moderate and vigorous-intensity activity levels (FLPA(1, 377) = 2.077, *p* = 0.150, ηp^2^ = 0.005; FMPA(1, 377) = 10.690, *p* < 0.050, ηp^2^ = 0.018 and FVPA(1, 377) = 17.323, *p* < 0.001, ηp^2^ = 0.044), whereas preventive health was only significant for light-intensity activities (FLPA(1, 377) = 9.068, *p* < 0.050, ηp^2^ = 0.023; FMPA(1, 377) = 1.952, *p* = 0.163, ηp^2^ = 0.005 and FVPA(1, 377) = 0.683, *p* = 0.409, ηp^2^ = 0.002).

A MANOVA with the listed barriers worse time allocation, motion disruption caused by COVID, physical effort, lack of family encouragement and sport environment) as independent variables and three levels of physical activity (light, moderate and vigorous) as dependent variables, revealed that worse time allocation due to the lockdown (F(3, 375) = 6.318, *p* < 0.001, ηp^2^ = 0.053, Wilks’ Lambda = 0.947), physical effort (F(3, 375) = 6.978, *p* < 0.001, ηp^2^ = 0.048, Wilks’ Lambda = 0.974) and lack of encouragement by family members (F(3, 375) = 3.390, *p* < 0.050, ηp^2^ = 0.026, Wilks’ Lambda = 1.000) were significant barriers for overall physical activity levels. Tests of Between-Subjects Effects indicated that physical effort (FLPA(1, 377) = 5.451, *p* < 0.050, ηp^2^ = 0.014; FMPA(1, 377) = 9.682, *p* < 0.050, ηp^2^ = 0.025 and FVPA(1, 377) = 9.895, *p* < 0.050, ηp^2^ = 0.026) and a worse time allocation due to the lockdown (FLPA(1, 377) = 13.043, *p* < 0.001, ηp^2^ = 0.033; FMPA(1, 377) = 11.534, *p* < 0.00, ηp^2^ = 0.030 and FVPA(1, 377) = 4.016, *p* < 0.050, ηp^2^ = 0.011) were significant barriers for all levels of physical activity, whereas lack of family encouragement was only a significant barrier for vigorous-intensity activities (FLPA(1, 377) = 2.317, *p* = 0.129, ηp^2^ = 0.006; FMPA(1, 377) = 0.007, *p* = 0.781, ηp^2^ = 0.000 and FVPA(1, 377) = 7.549, *p* < 0.050, ηp^2^ = 0.020).

### 3.4. Digital Support during Lockdown

For technological support, a one-sample *t*-test showed that respondents scored significantly higher on smartphone applications than the neutral usage frequency (value of three) (M = 3.31; SD = 1.669; t(396) = 3.699, *p* < 0.001). The use of wearables or smartwatches (M = 2.41; SD = 1.837; t(396) = −6.366, *p* < 0.001) and a combination of an application with a trackers or wearable (M = 2.27; SD = 1.738; t(392) = −8.302, *p* < 0.001) scored significantly lower than the neutral usage frequency (value of three). No significant difference was found for tracker usage (M = 2.90; SD = 1.785; t(396) = −1.125, *p* = 0.261).

All forms of mediated support scored significantly lower than the neutral usage frequency (value of three): online videos (M = 2.47; SD = 1.545; t(389) = −6.815, *p* < 0.001), posts on social media (M = 1.56; SD = 1.121; t(390) = −25.343, *p* < 0.001), groups or communities on social media (M = 21.58; SD = 1.090; t(388) = −25.685, *p* < 0.001), websites (M = 1.52; SD = 1.032; t(388) = −28.303, *p* < 0.001), lives on social media (M = 1.36; SD = 0.933; t(387) = −34.644, *p* < 0.001) and online sport lessons (M = 1.54; SD = 1.087; t(390) = −26.556, *p* < 0.001).

#### 3.4.1. Comparing Digital Support before and during Lockdown

Mean differences showed that, overall, the usage of technological support increased during the lockdown (see Figure 5). Paired sample *t*-tests indicated significant increases for use of smartphone applications (Mbefore = 2.72; SDbefore = 1.62 and Mduring = 3.31; SDduring = 1.67, t(396) = 9.314, *p* < 0.001), trackers (Mbefore = 2.62; SDbefore = 1.69 and Mduring = 2.90; SDduring = 1.79, t(396) = 5.059, *p* < 0.001) and smartphone applications in combination with a tracker or wearable (Mbefore = 2.07; SDbefore = 1.61 and Mduring = 2.27; SDduring = 1.74, t(392) = 4.204, *p* < 0.001) during the lockdown. Only the increase in the use of wearables or smartwatches was nonsignificant (Mbefore = 2.34; SDbefore = 1.78 and Mduring = 2.41; SDduring = 1.84, t(396) = 1.555, *p* = 0.121).

Paired sample *t*-tests indicated a significant increase in use of mediated support during the lockdown for all types of mediated support (see Figure 6): posts on social media (Mbefore = 1.35; SDbefore = 0.84 and Mduring = 1.56; SDduring = 1.12, t(390) = 5.695, *p* < 0.001), stories on social media (Mbefore = 1.19; SDbefore = 0.60 and Mduring = 1.36; SDduring = 0.93, t(387) = 5.636, *p* < 0.001), groups on social media (Mbefore = 1.30; SDbefore = 0.73 and Mduring = 1.58; SDduring = 1.09, t(388) = 6.243, *p* < 0.001), websites concerning exercise / sport (Mbefore = 1.28; SDbefore = 0.73 and Mduring = 1.51; SDduring = 1.03, t(387) = 5.938, *p* < 0.001), online videos (Mbefore = 1.84; SDbefore = 1.14 and Mduring = 2.46; SDduring = 1.55, t(388) = 9.905, *p* < 0.001) and online sport lessons (Mbefore = 1.14; SDbefore = 0.56 and Mduring = 1.53; SDduring = 1.08, t(389) = 7.870, *p* < 0.001).

#### 3.4.2. Digital Support per Level of Physical Activity during Lockdown

With regards to technological support, a MANOVA with the modes of technological support (smartphone application, tracker, wearable or smartwatch, combination) as independent variables and the three levels of physical activity (light, moderate and vigorous) as dependent variables, revealed that the use of smartphone applications (F(3, 351) = 3.257, *p* < 0.050, ηp^2^ = 0.027, Wilks’ Lambda = 0.973) and the use of trackers (F(3, 351) = 6.807, *p* < 0.001, ηp^2^ = 0.055, Wilks’ Lambda = 0.945) were significant predictors for all three levels of physical activity. Tests of Between-Subjects Effects indicated that smartphone applications were only significant predictors for vigorous-intensity physical activity levels (FLPA(1, 353) = 3.441, *p* = 0.063, ηp^2^ = 0.010; FMPA(1, 353) = 0.007 *p* = 0.933, ηp^2^ = 0.000 and FVPA(1, 353) = 5.502, *p* < 0.050, ηp^2^ = 0.015), whereas trackers were only significant predictors of light-intensity physical activity levels (FLPA(1, 353) = 19.258 *p* < 0.001, ηp^2^ = 0.052; FMPA(1, 353) = 3.471, *p* = 0.063, ηp^2^ = 0.010 and FVPA(1, 353) = 2.146, *p* = 0.144, ηp^2^ = 0.006).

In addition, a MANOVA with the modes of mediated support (online videos, posts on social media, groups on social media, websites, lives on social media, online sport lessons) as independent variables and the three levels of physical activity (light, moderate and vigorous) as dependent variables, revealed that the usage of websites was a significant predictor for all three levels of physical activity (F(3, 342) = 1.022, *p* < 0.050, ηp^2^ = 0.034, Wilks’ Lambda = 0.991). Tests of Between-Subjects Effects indicated that this significance was significant for light-intensity (F(1, 344) = 0.799, *p* = 0.372, ηp^2^ = 0.019) and moderate-intensity activities (F(1, 344) = 8.314, *p* < 0.050, ηp^2^ = 0.026), but not for vigorous -intensity activities (F(1, 344) = 6.762, *p* < 0.050, ηp^2^ = 0.002).

#### 3.4.3. Future Usage of Digital Support

Regarding the intended future use of technological support (see Figure 7), a one-sample *t*-test showed that respondents scored significantly higher on their intention to use smartphone applications (M = 3.98; SD = 1.447;t(369) = 13.040, *p* < 0.001), trackers (M = 3.70; SD = 1.607;t(323) = 7.812, *p* < 0.001) and wearables (M = 3.32; SD = 1.774;t(295) = 3.080, *p* < 0.001) in the future than compared to the neutral probability rate (value of three). No significant difference was found for their intention to keep using combinations of an application and a tracker or wearable in the future, compared to the neutral probability rate (value of three) (M = 3.08; SD = 1.771;t(293) = 0.757, *p* = 0.449).

A one-sample *t*-test showed that all forms of mediated support scored significantly lower than the neutral probability rate (value of three) on their intention to keep using these supports in the future, with the exception of online videos (see Figure 7),: posts on social media (M = 1.94; SD = 1.357;t(287) = −13.283, *p* < 0.001), lives on social media (M = 1.66; SD = 1.170;t 275) = −19.091, *p* < 0.001), groups or communities on social media (M = 1.91; SD = 1.277;t 285) = −14.400, *p* < 0.001), websites (M = 2.04; SD = 1.363; t(283) = −11.845, *p* < 0.001), online sport lessons (M = 1.78; SD = 1.114; t(249) = −18.820, *p* < 0.001) and online videos (M = 3.06; SD = 1.580; t(346) = 0.714, *p* = 0.476).

### 3.5. Affect during Lockdown

One-sample *t*-tests showed that respondents felt significantly less of all affective measures compared to the neutral agreement score (value of three): well-being (M = 2.33; SD = 3.99; t(424) = −3.471, *p* < 0.001); fear of COVID (M = 2.11, SD = 0.65; t(425) = −28.343, *p* < 0.001), anxiety (M = 2.14, SD = 0.89; t(426) = −19.998, *p* < 0.001) and fatigue (M = 2.51, SD = 1.01; t(426) = −10.096, *p* < 0.001). Using a one-sample *t*-test comparing the scores of the international sample used to develop the short PANAS scale (2011), respondents of this survey experienced significantly less negative affect (M international = 12.35 and Msample = 10.99; SD = 3.73, t(425) = −7.528, *p* < 0.001) and significantly less positive affect (M international = 19.09 and Msample = 13.31; SD = 3.18, t(424) = −37.51, *p* < 0.001) than the international average.

#### Affect per Level of Physical Activity during Lockdown

With regards to affective measures, MANOVA with the affective measures (well-being, fear of COVID, anxiety and fatigue) as independent variables and the three levels of physical activity (light, moderate and vigorous) as dependent variables revealed that positive affect (F(3, 378) = 7.663, *p* < 0.001, Wilks’ Lambda = 0.943, ηp^2^ = 0.057 and fear of COVID (F(3, 378) = 8.226 *p* < 0.001, ηp^2^ = 0.061, Wilks’ Lambda = 0.939) were significant predictors for all three levels of physical activity. Tests of Between-Subjects Effects indicated that a positive affect was a significant predictor for all levels of physical activity (FLPA(1, 380) = 12.920, *p* < 0.001, ηp^2^ = 0.033; FMPA(1, 380) = 7.142 *p* < 0.050, ηp^2^ = 0.018 and FVPA(1, 380) = 10.567, *p* < 0.001, ηp^2^ = 0.027), whereas fear of COVID was only a significant predictor for vigorous-intensity physical activity levels (FLPA(1, 380) = 2.353 *p* = 0.189, ηp^2^ = 0.006; FMPA(1, 380) = 3.294, *p* = 0.070, ηp^2^ = 0.012 and FVPA(1, 380) = 4.270, *p* < 0.050, ηp^2^ = 0.038).

### 3.6. Sufficiently Active or Insufficiently Active during Lockdown?

In order to better understand the predictors of sufficiently active individuals, a distinction between insufficiently active and sufficiently active individuals during the lockdown was made. According to the World Health Organization, individuals are considered sufficiently active when they engage in at least 150 min of moderate-intensity or 75 min of vigorous-intensity exercise per week [44]. Overall, it is recommended to engage in light intensity activities as much as possible, but recommendations are not specified with further details [44]. In this sample, 53.4% of respondents met these requirements—a similar number to the national average (adults from 18 to 77 years old) found by Health Interview Survey Interactive Analysis in 2018: 58.4% [54]. Out of those, 32.50% were sufficiently active exclusively due to their engagement in moderate exercise and 41% were classified as such exclusively due to their performance of vigorous exercise.

#### 3.6.1. Physical Activity during Lockdown

On average, the sufficiently active group engaged more frequently and for more minutes in physical activity of all three levels of intensity (see Table 2). Levene’s test showed that the variances for frequency (based on mean) in moderate-intensity exercise (F(1, 386) = 20.255, *p* < 0.001) and in vigorous-intensity exercise (F(1, 386) = 76.095, *p* < 0.001) were not equal and variances for frequency (based on mean) in light-intensity exercise (F(1, 386) = 0.384, *p* = 0.791) were equal. In addition, Levene’s test also showed that the variances for duration (based on mean) moderate-intensity exercise (F(1, 386) = 4.171, *p* < 0.050) and in vigorous-intensity exercise (F(1, 386) = 35.886, *p* < 0.001) were not equal and variances for duration (based on mean) in light-intensity exercise ((F(1, 386) = 0.384, *p* = 0.536) were equal. A non-significant *p*-value of Levene’s test show that the variances are indeed equal and there is no difference in variances of both groups. Therefore, A MANOVA was only conducted for exercise frequency and duration considering moderate- and vigorous-intensity exercise. A MANOVA revealed that, overall, the difference across all four dependent variables (frequency and duration at two levels of exercise) were significant (F(4, 391) = 128.554, *p* < 0.001, ηp^2^ = 0.568, Wilks’ Lambda = 0.432). Post hoc tests showed that the differences between the two groups in terms of exercise frequency were significant for both moderate-intensity (F(1, 396) = 85.557, *p* < 0.001, ηp^2^ = 0.178) and vigorous-intensity physical activity (F(1, 396) = 227.043, *p* < 0.001, ηp^2^ = 0.366). In addition, the differences between the two groups in terms of exercise duration were also significant for both moderate-intensity (F(1, 394) = 57.525, *p* < 0.001, ηp^2^ = 0.127) and vigorous-intensity physical activity (F(1, 394) = 181.174, *p* < 0.001, ηp^2^ = 0.315). Per week, on average, the insufficiently active group engaged for 167.76 min in light (SD = 159.84), 52.36 min in moderate (SD = 45.23) and 11.75 min in vigorous exercise (SD = 21.48), whereas the sufficiently active group engaged for 229.36 (SD = 181.97), 179.87 (SD = 141.43) and 133.50 (SD = 113.02) in light, moderate and vigorous exercise per week, respectively.

#### 3.6.2. Comparing Physical Activity Levels before and during Lockdown

One sample *t*-tests indicated that the insufficiently active group perceived their change in vigorous-intensity activities as significantly lower (M = 80.24; SD = 50.29; t(167) = −5.093; *p* < 0.001) than the neutral value of 100 (which was indicated as ‘no change in comparison with before the lockdown’). No significant differences were found for light-intensity activities (M = 105.60; SD = 52.84; t(167) = 1.372; *p* = 0.172) and moderate-intensity activities (M = 96.79; SD = 55.90; t(167) = –0.745, *p* = 0.457).

For the sufficiently active group, one sample *t*-tests revealed that their perceived change for light-intensity (M = 118.95; SD = 46.39; t(227) = 6.167, *p* < 0.001), moderate-intensity (M = 117.72; SD = 47.44; t(227) = 5.640, *p* < 0.001) and vigorous-intensity activities (M = 118.42; SD = 57.38; t(227) = 4.847, *p* < 0.001) was significantly higher than the neutral value of 100 (no change).

Levene’s test showed that the variances for perceived change in physical activity during lockdown (based on mean) for light-intensity exercise (F(1, 394) = 4.847, *p* < 0.050), moderate-intensity exercise (F(1, 394) = 4.5867, *p* < 0.050) and vigorous-intensity exercise (F(1, 386) = 7.417, *p* < 0.050) were not equal. A MANOVA was performed and revealed that the differences in perceived change between the sufficiently active and insufficiently active group were significant (F(3, 392) = 17.908, *p* < 0.001, ηp^2^ = 0.212, Wilks’ Lambda = 0.879). In addition, post hoc tests indicated that these differences were significant for all three levels of physical activity (F_LPA_(1, 394) = 7.115, *p* < 0.050, ηp^2^ = 0.18; F_MPA_(1, 394) = 16.172, *p* < 0.001, ηp^2^ = 0.039; F_VPA_(1, 394) = 47.497, *p* < 0.001, ηp^2^ = 0.108).

#### 3.6.3. Predictors of Belonging to the More Active Group

A forward logistic regression analysis (likelihood-ratio) to investigate what predicted if respondents belonged to the insufficiently or sufficiently active group was conducted (0 = belonging to the insufficiently active group; 1 = belonging to the sufficiently active group). The dependent variable was a dichotomous variable with scores 1 = belonging to the sufficiently active group and 0 = belonging to the insufficiently active group. A number of 336 cases or 78.9% of the sample was included, with 91 cases of the sample missing. The unstandardized beta weight for the constant; B = −2.960, SE = 0.763, Wald = 15.029, *p* < 0.001. Five variables were found to predict belonging to the sufficiently active group: experiencing time allocation as an opportunity, physical performance as a benefit, using an application in combination with a tracker before the lockdown, watching online videos during the lockdown and not experiencing physical effort as a barrier. The estimated odds ratio favoured an increase of 6.53% [Exp (B) = 1.348; 95% CI (1.112, 1.634)] every one unit increase of time allocation as an opportunity, an increase of 8.54% [Exp (B) = 1.802; 95% CI (1.355, 2.398)] every one unit increase of physical performance, an increase of 6.46% [Exp (B) = 1.333; 95% CI (1.120, 1,587)] every one unit increase of using an application in combination with a tracker before the lockdown, an increase of 5.85% [Exp (B) = 1.199; 95% CI (1.017, 1.414)] every one unit increase of watching online videos during the lockdown, and a decrease of 3.37% [Exp (B) = 0.673; 95% CI (.540, 0.838)] every one unit increase of physical effort of chances to belonging to the sufficiently active group. Taking all these independent variables together resulted in an increase of 11.94% for belonging to the more active group.

## 4. Discussion

The aim of the present study was to examine how Belgian citizens adapted their exercise routines during the lockdown. This paper found that, overall, respondents in our survey saw the lockdown as an opportunity to increase exercise behaviours. Results, however, also revealed that the lockdown widened the gap between insufficiently active and sufficiently active individuals even more. This means that, overall, sufficiently active individuals exercised more, while insufficiently active individuals exercised an equal amount (or sometimes even less).

### 4.1. Physical Activity

Respondents of this sample reported to engage most frequently and with the highest duration in light-intensity physical activity, followed by moderate and vigorous-intensity activity. Overall, they perceived their level of engagement in all three intensities of physical activity as higher than before the lockdown, although this perception only showed to be significant for light and moderate intensity-activities. This is in line with the research of Scheerder and colleagues [55] that also found that a majority of the Belgian population perceived to exercise as much or more than before the lockdown.

### 4.2. Benefits and Barriers

Considering the benefits related to physical activity and the lockdown, our findings revealed that, overall, all listed benefits were accepted by the sample with the exception of social interactions, such as being able to meet friends when exercising. This finding is surprising as social interactions (meeting one or two friends outside of the household) was allowed only during outdoor exercise [56], and, in another research work, individuals indicated to miss playing sports with others [55]. However, in the research report of Fancourt and colleagues [57], it is also stated that individuals generally avoid meeting friends, neighbours and families during a pandemic, with an exception for individuals who live alone. An explanation for this avoidance can be that individuals have great concerns for the health of their loved ones during the pandemic [58]. It is expected that, when the phases to return back to ‘normal’ will be implemented [59,60], social interactions will gradually be considered as a prominent benefit again.

The perception of a *better time allocation* due to the lockdown was found to be a significant predictor for the engagement in all three levels of physical activities. This finding was similar to the results of Scheerder and colleagues [55], where more than half of the people indicated having more time due to the implemented corona measures, and that this greater availability of time translated into more effective sports and exercise behaviours. The perception of a better time allocation can be explained by the limited outings that were allowed because of the lockdown, including for commuting purposes [3].

Physical performance predicted engagement in moderate-intensity and vigorous intensity activities, whereas preventive health was found to predict light-intensity activities. Previous research concerning preference for exercise intensity found that a performance-approach goal (i.e., similar to the physical performance as a benefit in the present study) mediated preference for more intense-exercise physical activity participation [61]. These results were only found to be true for men. However, our sample consists mostly of female respondents. We recommend future research to look more closely at how physical performance is perceived differently by men versus women.

As for *preventive health* predicting light-intensity exercise, it seems that people associate light-intensity activities with better health because doing ‘something’ is healthier than doing nothing. This might be related to the encouragements of the Belgian government to keep moving during the pandemic, emphasizing that every step counts [62]. Home exercise and walking—framed as two light-intensity activities—are promoted as healthy [63]. In addition, the Flemish ‘Taking 10,000 Steps a Day’ campaign also promotes light-intensity activities for the benefit of good health: “You can still get plenty of exercise, even if you are not yet ready to exercise” [63].

Most listed barriers to engaging in physical activity were not experienced by the respondents in our study. However, a *worsened time allocation* due to the lockdown and *physical effort* were found to be significant barriers to engage in all three levels of physical activity. This is in line with lack of time being often cited as a barrier to engage in physical activity [27,64]. Furthermore, perception of effort refers to the conscious sensation of how hard, heavy, and strenuous a physical task is [65]. As physical activity is an effortful activity that requires energy [66], it is likely to be perceived as a high cost-and people tend to prefer engaging in activities that demand less effort [67].

In addition, lack of family encouragement was found to be a barrier to engage in vigorous-intensity activities. Overall, social support is an important correlate to overall health [68] and, according to the theory of planned behaviour, subjective norms contribute to health behaviours [69]. Previous research found that parent and peer encouragement of vigorous-intensity physical activity increased involvement in activity among college students [70]. These results seem to extend this need for support from parents to partners. Further research should explore more deeply why encouragement by loved ones is extra relevant for high-intensity activities.

### 4.3. Digital Support

In general, the use of technological support significantly increased during the lockdown for all types of support, except for wearables/smartwatches. Additionally, respondents in our study indicated using trackers occasionally and smartphone applications significantly more than occasionally during the lockdown. Similar results were found by Stragiers and colleagues [71], who investigated the *smartphone application* Strava and the *activity tracker* Polar. They found that Strava was opened and used more to record or synchronize activities, but also to comment more on other users during the Belgian lockdown. Concerning Polar, the tracker recorded 41.9% more sessions in its most popular sports in April than in February. Future research should investigate what the added value could be of a wearable or smartwatch, as individuals appear to be satisfied with the use of smartphone applications. In addition, smartphone applications were found to be a significant predictor for engagement in vigorous-intensity activities, whereas *trackers* were found to be significant predictors of participation in light-intensity exercise. Regarding smartphone applications, the systematic review of Feter and colleagues [72] found that smartphone applications help in the promotion of physical activity by increasing exercise minutes and steps, but no distinction was made per physical activity level. With regards to trackers, the systematic review and meta-analysis of Brickwood and colleagues [73] found that trackers increased daily step (light-intensity physical activities) count as well as moderate-intensity and vigorous-intensity activities. With these findings in mind, future research should investigate what types of technological support are most accommodating for which levels of physical activity and why.

Considering mediated support, all types of support were used significantly more during the lockdown compared to before, especially *online videos*. However, all different forms of mediated support were still used less than occasionally during the lockdown. Therefore, it seems that the trend to move life to the online sphere has been picked up for physical activities, albeit only to a limited extent. A possible explanation for this is that the online trend is still very new—which boomed due to the coronavirus [30]—and people are still getting used to it. This is comparable to the results of research on cooking habits during the corona pandemic (conducted in Belgium and the Netherlands) that indicated a willingness of people to try to interact online with family and friends during meals, but 32.6% of those who tried still found it somewhat uncomfortable [74]. Another explanation could be that mediated support is still associated with fitspiration content, which has been found to elicit negative feelings in some users [75], likely limiting the demand for this content. Future research could investigate the adherence to mediated support for exercising in the long run. In addition, the higher usage of online videos relates to the consistent findings on the popularity of virtual personal trainers [76,77]. Furthermore, the usage of websites was found to be a significant predictor for engagement in all levels of physical activities. This might be explained by previous research conducted by Berry and colleagues [78], which found that individuals mostly look for information about physical activity online, whereby the goal to be physically active was a significant factor in the odds of looking for information on the internet.

### 4.4. Affect

Concerning affect, the sample indicated to feel significantly less anxiety, fear of COVID, fatigue, positive and negative affect than feeling neutral. Positive affect was found to be a significant predictor for all levels of physical activity, whereas fear of COVID proved to be a significant predictor of vigorous-intensity activities. Previous research has frequently associated positive affect and physical activity as a reinforcing spiral: people who experience positive affect exercise more, and people who exercise more experience more positive affect [79,80,81,82]. Furthermore, physical activity has also been found to be used as a coping mechanism against stressors, especially at higher-intensity levels [83,84]. However, the specific link between vigorous-intensity physical activity and fear of COVID is yet to be explored.

### 4.5. Sufficiently Active vs. Insufficiently Active

As previous research shows that motivations and barriers differ between active and inactive individuals [45], this critical distinction was also made in this study. A comparison between individuals who met the guidelines prescribed by the World Health Organization and those who did not meet these guidelines showed a significant difference on all three levels of intensity in favour of the sufficiently active group. On average, the sufficiently active group engaged significantly more in physical activity per week than the insufficiently active group. The insufficiently active group indicated to perceive a significant decrease in vigorous-intensity activities during the lockdown in comparison with before, but no significant changes in their physical activity levels for light-intensity and moderate-intensity activity were found. However, the sufficiently active group indicated to perceive a significant increase during the lockdown for all three levels of physical activity.

These results are in contrast to the findings of Constandt and colleagues [21], who found that both sufficiently (36%) and insufficiently (58%) active individuals increased their exercise level during the lockdown in Belgium. An explanation for this can be found in the difference in the timings of the two online surveys. According to the model of physical exercise and habit formation [85], individuals might have felt the need to exercise (for their health [18,19]), felt social desirability to exercise (encouragement by governments [4,5]), had good intentions, and eventually exercised at the beginning of the lockdown. However, maintaining regular exercise is difficult and they might have quit, for example, by creating personal goals that were too difficult to achieve—this appears to be what happened with the insufficiently active group: they might have increased their exercise behaviours at the beginning of the lockdown but eventually quit towards the end of the lockdown [85].

### 4.6. Predictors of Belonging to the More Active Group

As our findings show that the implemented lockdown has widened the gap between the insufficiently and sufficiently active individuals even more, we wanted to understand what contributed to belonging to the sufficiently active group during the lockdown. The first significant predictor of belonging to the sufficiently active group was found to be already *relying on an application in combination with a tracker or wearable as support* to exercise before the lockdown. This indicates that they were already involved with physical activity —likely seeing exercise as part of their identities—in contrast to the insufficiently active group [86]. Furthermore, previous research shows that people who use applications in combination with a tracker or wearable are generally more active than people who do not use these combinations [87]. For this reason, future research should focus on adaptations that wearable technologies can make to attract not only sufficiently but also insufficiently active individuals.

For example, *not focusing on physical performance goals* was the second predictor for belonging to the more active group. As physical performance is valued by the more active group, stimulating them to reach their goals is found to be effective [88]; however, this might work as aversive for the insufficiently active group as they do not (yet) value physical performance. This might be due to having lower self-efficacy in reaching these goals [89] or not perceiving it as a benefit, as they do not experience bodily improvements yet. Future research should look into different strategies, besides focusing on physical performance, to increase physical activity levels, such as by entertaining them with activities that indirectly benefit physical performance [90].

The third predictor was *better time allocation* due to the lockdown. Experiencing a better time allocation can be linked to the interpretation that the lockdown is an opportunity to organize time differently (having the perception of more time). This is plausible to relate to the associations that can be made between lockdown and boredom [39]. Commonly used strategies to cope with boredom are reading, daydreaming, socializing, watching television, physical activity, and trying something new [91]. Given that the government has limited non-essential outings, but has authorized, and even incentivized, outings for exercise, it is likely that people are relying on physical activity as a coping mechanism for boredom. Individuals that consider physical activity enjoyable and pleasurable-in this case, the sufficiently active group, are even more likely to resort to this activity as a boredom coping strategy [91], whereas the insufficiently active group likely engaged in other activities they enjoyed more than physical activity.

The latter idea is in line with the fact that the sufficiently active group does *not experience physical activities as a physical effort* (the fourth predictor), whereas the insufficiently active group does [67]. Future research should focus on identifying what individuals consider to be ‘a good time allocation’ to engage in physical activities and how they feel they can fit this time into their schedules experiencing minimum amounts of effort. Lastly, *watching online videos during the lockdown* contributed to belonging to the more active group. The use of online videos has been found to be popular and engaging and can be highly personalized if you know where and how to look for this type of guidance [92]. It is plausible that the insufficiently active group had either no interest in looking for this content [93] or was exposed to an overload when searching, which has been found to have adverse effects on decision-making [94]. Future research should focus on identifying how to make online videos more accessible and appealing for insufficiently active individuals.

## 5. Practical Implications

Overall, the five predictors that are explained shed light on the importance of creating habits to sustain a healthy lifestyle, even during stressful and uncertain periods. In addition, the presented insights that can inform policies and interventions to support physical activity engagement, even in times of social isolation and uncertainties.

For individuals who really do not have the time to exercise, increasing physical movement while performing daily activities is likely to be the easiest way to increase engagement in physical activity. With the help of technology, individuals can be reminded of their inactive behaviours and can be nudged towards a more active lifestyle by setting daily step-goals and reminding them in their daily activities to take more steps [95]—for example, stimulating users to put their bottle of water in the kitchen instead of on their desk to get steps every time they drink water. Governments could design billboards pointing out that you should take the stairs instead of the elevator in public places or park your car a little bit further from your destination.

However, for most people, it might be a matter of increasing their perceived self-efficacy. Hence, interventions and policies aimed at increasing self-efficacy should remind individuals that exercise performed in any intensity and duration can confer health benefits [96]. For example, wearables or interventions on the work floor that remind (e.g., for example, by playing a certain sound) individuals, who inhibit a sedentary lifestyle, that standing up every now and just walking a few steps can already lead to health benefits, might help remove the idea that they should be able to run a marathon to enjoy the benefits of physical activity.

The perception of physical effort as a barrier to exercise for the insufficiently active group supports the claim that they experience some level of displeasure during exercise; therefore, focusing on enjoyment and benefits of exercise could be beneficial for this group. In addition, finding an activity that the individual enjoys can help with this matter. A suggestion could be that sports centres create initiatives for adults where they can participate in different sports to discover what they like and enjoy, rather than having to get involved right away in an unfamiliar and maybe (in their opinion) unpleasant sport. In this way, they could already engage more in physical activities during their selection and persisting with the chosen activity being made easier. Consequently, this will affect agency, which is described in the literature as an important intrinsic motivator for physical activity [97,98]. An interesting platform that takes several of these aspects into account is Gympass, which focuses on on-demand workouts to fit your schedule and monthly memberships [99].

## 6. Limitations and Strengths

A first important limitation for this study is the reliance on self-reported measures, although it is still the most widely used type of physical activity measurement [100]. Individuals had estimate their physical activity behaviours, their use of digital technologies, and their affect, which can also be measured by technological devices. However, the measures introduced by the lockdown only allowed for an online questionnaire, subsequently relying on self-reported measures. Therefore, people might have overestimated or underestimated their time spent on exercise, as a result of which belonging to one or the other group cannot be substantiated by explicate figures. A second limitation is that we worked with a convenience sample, which means that individuals self-selected if they filled in the questionnaire or not. Therefore, results cannot be generalized for a certain population; instead, coherence was examined. Additionally, this might explain the overall active sample. A third limitation is that the questionnaire mainly consisted of closed items, which may have restricted the factors that strengthen or limit physical activity engagement. Nevertheless, each question provided respondents with the opportunity to indicate ‘other’ and fill in what they perceived was missing in the suggested items. It may still be plausible that there are other factors contributing to limited engagement in physical activity that were not captured in this study, such as lack of self-motivation, boredom with exercise, low self-efficacy, being injured but also needing to take care of children, and lack of self-management skills during the lockdown [64,101].

Although prior research has already analysed physical activity behaviours during the first lockdown, this study collected data after almost ten weeks of lockdown, allowing for capturing a better picture of how individuals adapted their physical activity behaviours to the lockdown—rather than capturing a first glance reaction to reduced mobility. In addition, this paper also looks further than physical activity alone (taking into account benefits, barriers, technological support tools, mediated support tools, and affect) and takes analyses beyond descriptive analysis. Thus, the findings of this paper allow for understanding physical activity behaviours during the lockdown at a more in-depth level, including explaining factors as to why individuals belonged to the more active group during the lockdown. Additionally, this paper explains how understanding these factors will help governments and institutions to guide insufficiently active individuals in increasing their physical activity levels in general and during future lockdowns.

## 7. Conclusions

Individuals who participated in this study generally exercised more than before the lockdown, especially in light-intensity and moderate-intensity exercise. When a distinction in our sample was made between individuals who were sufficiently active and insufficiently active before the lockdown, it was found that sufficiently active individuals engaged even more in physical activity, whereas insufficiently active individuals exercised equally (i.e., not enough) or even less compared to before the lockdown. This paper elaborated on five factors that explained why individuals belonged to the more or less active group. This study was limited by the use of self-reported measures. Notwithstanding this limitation, the study suggests that individuals should be consistent with their exercise routines for successful maintenance or even increase in physical activity levels during uncertain times like a lockdown. Subsequently, governments and institutions should be aware of the fact that the most vulnerable group with regards to physical activity is even in more need of stimulation and support when experiencing hard times. A lockdown is one example that can be expanded to other stressful times in an individual’s life. Highlighting that even light-intensity physical activities can be performed at any time and duration, can be a solution for this matter.

## Figures and Tables

**Figure 1 ijerph-18-05555-f001:**
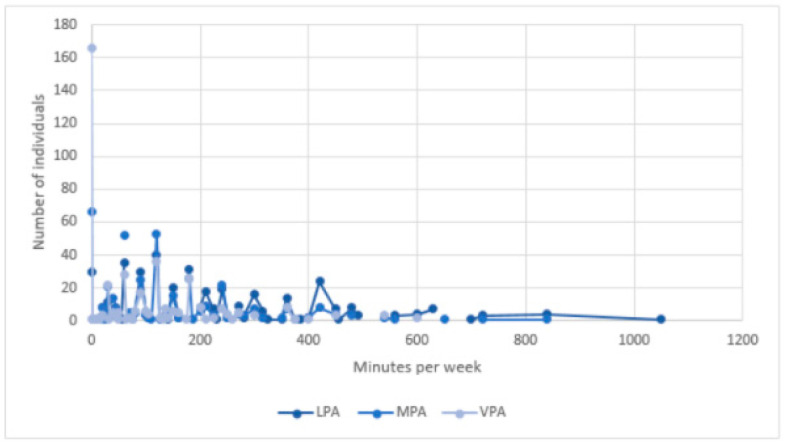
Frequency per week (per physical intensity level.

**Figure 2 ijerph-18-05555-f002:**
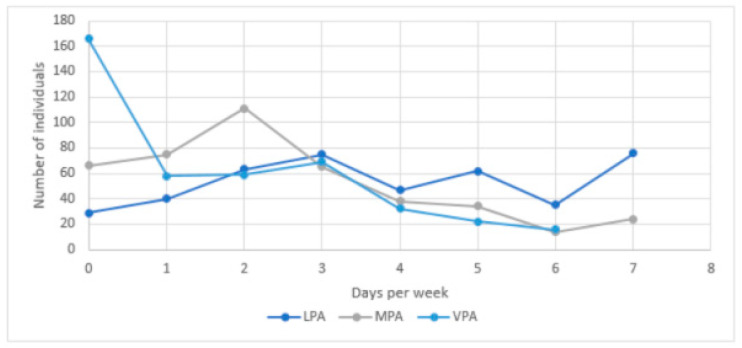
Duration of exercise per session (per physical intensity level).

**Figure 3 ijerph-18-05555-f003:**
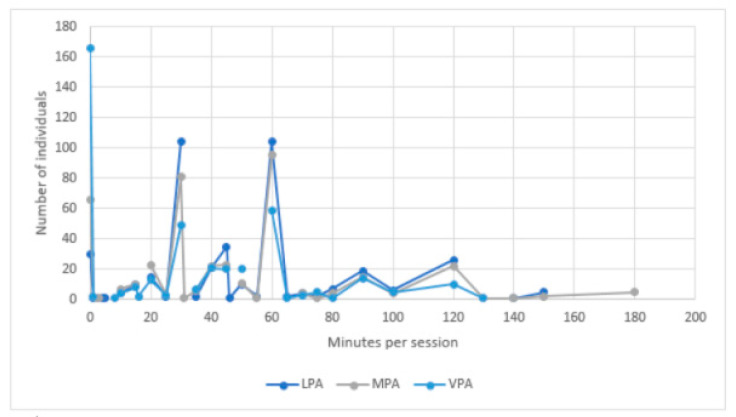
Overall level of physical activity per week (per physical intensity level).

**Figure 4 ijerph-18-05555-f004:**
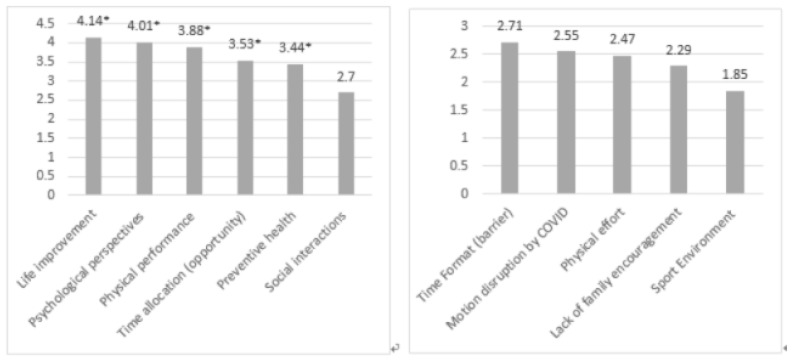
Mean scores of benefits and barriers (5-point Likert Scale). * = significantly higher score than the neutral value of three.

**Figure 5 ijerph-18-05555-f005:**
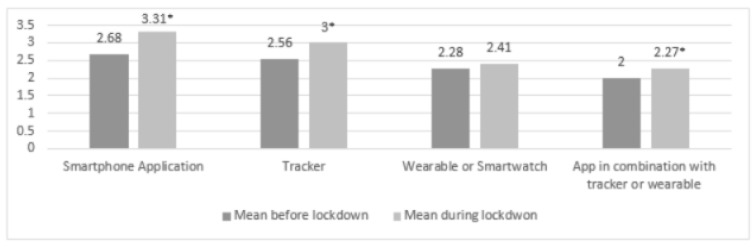
Mean scores of technological supports (5-point Likert Scale). * = significant increase in comparison with before the lockdown.

**Figure 6 ijerph-18-05555-f006:**
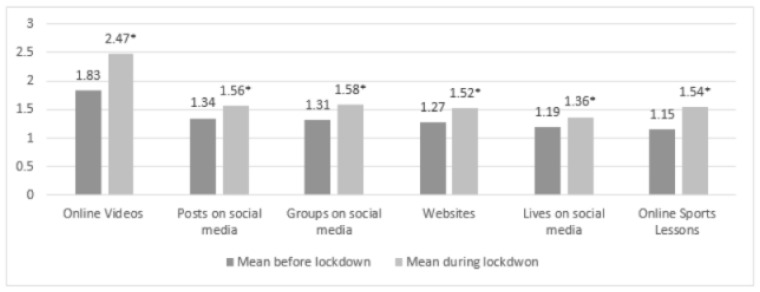
Mean scores of mediated supports (5-point Likert Scale). * = significant increase in comparison with before the lockdown.

**Figure 7 ijerph-18-05555-f007:**
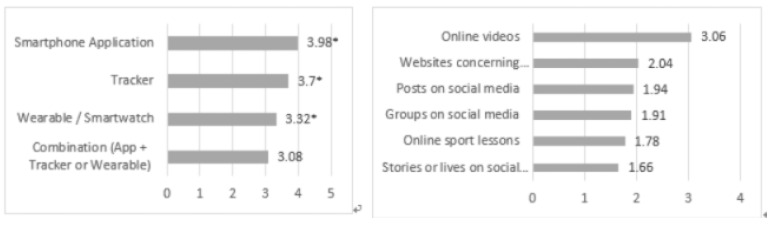
Mean scores of the intended future use of digital support (5-point Likert Scale). * = significantly higher score than the neutral value of three.

**Table 1 ijerph-18-05555-t001:** Characteristics of respondents.

Continuous Study Variables	M	SD
Age (years)	34.00	14.12
**Categorical Study Variables**	**n**	**%**
Gender		
Men	354	17.1
Women	73	82.9
Education		
Elementary Education	7	1.6
Secondary Education	48	11.2
Bachelor’s degree	182	42.6
Master’s degree	149	34.9
PhD	41	9.6
Living Situation		
With Parents	124	29.0
With Partner	117	27.4
With Partner and Children	58	13.6
Alone	50	11.7
Other	78	18.3%
Number of Children in the Household		
No children	326	76.3
One child	25	5.9
Two children	52	12.2
More than two children	24	5.6
Change in Household due to Coronavirus		
Yes	54	12.6
No	373	87.4
Employment Situation		
Full-Time Employment	174	40.7
More than Part-Time Employment (>50%)	47	11.0
Part-Time Employment (=50%)	15	3.5
Less than Part-Time Employment (<50%)	10	2.3
Not Employed	61	14.3
Student	120	28.1
Change in Employment		
Yes	73	17.1
No	354	82.9

**Table 2 ijerph-18-05555-t002:** Insufficiently active group vs. sufficiently active group during lockdown.

		Insufficiently Active Group			Sufficiently Active Group		
		Mean	SD	Median	Mean	SD	Median
LPA	Duration (min)	43.69	31.43	40.00	51.33	29.14	50.00
	Frequency (days/week)	3.48	2.22	3.00	4.11	2.15	4.00
MPA	Duration (min)	29.09	27.93	30.00	54.68	36.56	50.00
	Frequency (days/week)	1.54	1.34	1.00	3.20	1.98	3.00
VPA	Duration (min)	8.96	17.56	0.00	44.01	30.20	45.00
	Frequency (days/week)	0.43	0.87	0.00	2.62	1.73	3.00

## Data Availability

The data presented in this study are available on request from the corresponding author. The data are not publicly available due to respondents only agreeing to share responses with the contributing researchers, corresponding to the ethical guidelines applied by the University of Antwerp.

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
