# Peer review of "Physical Activity during the First Lockdown of the COVID-19 Pandemic: Investigating the Reliance on Digital Technologies, Perceived Benefits, Barriers and the Impact of Affect"

_ijerph, 2021, doi:10.3390/ijerph18115555_

Round 1

Reviewer 1 Report

Dear Authors:

After review the article "1153695"

The next changes are presented in order to consider

Attached the inform

In Advanced

King Regards

Author Response

Dear Reviewer,

The co-authors and myself would like to thank you to take the time reading our submitted manuscript and providing us with the necessary feedback to improve our submission.

Please see the attachment with our answers to your valuable feedback. 

Kind regards, 

the authors 

Reviewer 2 Report

Dear Authors,

I was reading your manuscript entitled „Physical activity during the first lockdown of the COVID-19  2 pandemic: investigating the reliance on digital technologies,  3 perceived benefits, barriers and the impact of affect”. The topic you chose is very actual and important. Physical inactivity is one of the leading problems and one of the factors that lead to overweight and obesity. The COVID pandemic without any doubt limits the possibility to exercise and forces many people to stay at home.  Your research raised a very important problem, but it requires a major revision before acceptance. I can recommend your manuscript for publication only after major changes, below you will find a detailed description. 

Introduction
Authors should give more information about physical activity (PA), there is a description of the situation and many restrictions in COVID pandemic but on the other hand, there is a lack of explanation about the amount of recommended physical activity, the benefits of PA

authors mentioned about three categories of PA - there is a lack of information about them, what was the method of calculation?

the text should be spread into multiple paragraphs to improve readability.

authors should write the recommendation of PA for adult people

line 87-100 – this should go into the methodology section
there was no clear aim of the study

Materials and methods
Line 115-127 – this part should be described in the Results section, not in the subsection sample and procedure

there is a lack of information about tools used in the project (a detailed description of that tools the questionnaires and the mobile application for calculating the PA - duration and time),

there was no information about statistical analysis
Results and Discussion

That sections should be a rethink and revised once again. There is a lack of clarity. Some of the results were put and described in the methodology section, there is a lack of explanation of how those PA categories were calculated what were the units of PA was it only a time duration or should it be also Met-min/week?

what are the other factors which can limit PA

Conclusion
authors should not describe the aim in the conclusion

the conclusion should be rephrased

Authors should add the limitation of the study - what are the limitation of the on-line questionnaires

Author Response

(The authors gave the same response as above.)

Round 2

Reviewer 1 Report

Dear aurthors

Thanks for condifer our suggestions

Now, the article is ready

In Advanced

King Regards